

# Population structure and selective signature of Kirghiz sheep by Illumina Ovine SNP50 BeadChip

Ruizhi Yang[1], Zhipeng Han[2,3], Wen Zhou[2,3], Xuejiao Li[2], Xuechen Zhang[2,3], Lijun Zhu[2,3], Jieru Wang[1], Xiaopeng Li[2], Cheng-long Zhang[2], Yahui Han[2], Lianrui Li[1,2,3,4] and Shudong Liu[2,3,4]

[1] College of Life Science and Technology, Tarim University, Alar, Xinjiang, China
[2] College of Animal Science and Technology, Tarim University, Alar, Xinjiang, China
[3] Xinjiang Production and Construction Corps, Key Laboratory of Tarim Animal Husbandry Science and Technology, Alar, Xinjiang, China
[4] Xinjiang Production and Construction Corps, Engineering Laboratory of Tarim Animal Diseases Diagnosis and Control, Alar, Xinjiang, China

Corresponding authors
Lianrui Li, lilianrui51@163.com
Shudong Liu, liushudong63@126.com

## ABSTRACT

**Objective**. By assessing the genetic diversity and associated selective traits of Kirghiz sheep (KIR), we aim to uncover the mechanisms that contribute to sheep's adaptability to the Pamir Plateau environment.

**Methods**. This study utilized Illumina Ovine SNP50 BeadChip data from KIR residing in the Pamir Plateau, Qira Black sheep (QBS) inhabiting the Taklamakan Desert, and commonly introduced breeds including Dorper sheep (DOR), Suffolk sheep (SUF), and Hu sheep (HU). The data was analyzed using principal component analysis, phylogenetic analysis, population admixture analysis, kinship matrix analysis, linkage disequilibrium analysis, and selective signature analysis. We employed four methods for selective signature analysis: fixation index (Fst), cross-population extended homozygosity (XP-EHH), integrated haplotype score (iHS), and nucleotide diversity (Pi). These methods aim to uncover the genetic mechanisms underlying the germplasm resources of Kirghiz sheep, enhance their production traits, and explore their adaptation to challenging environmental conditions.

**Results**. The test results unveiled potential selective signals associated with adaptive traits and growth characteristics in sheep under harsh environmental conditions, and annotated the corresponding genes accordingly. These genes encompass various functionalities such as adaptations associated with plateau, cold, and arid environment (ETAA1, UBE3D, TLE4, NXPH1, MAT2B, PPARGC1A, VEGFA, TBX15 and PLXNA4), wool traits (LMO3, TRPS1, EPHA5), body size traits (PLXNA2, EFNA5), reproductive traits (PPP3CA, PDHA2, NTRK2), and immunity (GATA3).

**Conclusion**. Our study identified candidate genes associated with the production traits and adaptation to the harsh environment of the Pamir Plateau in Kirghiz sheep. These findings provide valuable resources for local sheep breeding programs. The objective of this study is to offer valuable insights for the sustainable development of the Kirghiz sheep industry.

## INTRODUCTION

Sheep are one of the earliest domesticated animal species, with an estimated history of domestication dating back approximately 11,000 years (*Morell Miranda, Soares & Günther, 2023*). During the Neolithic era, sheep emerged as one of the most successful domesticated animals, providing humans with a diverse range of resources including meat, dairy products, lambskins, and wool (*Alberto et al., 2018*). Nevertheless, due to different natural environments and various domestication methods, various sheep breeds show significant differences in morphology, physiology, behavior, and their ability to adapt to the environment (*Kalds et al., 2022*).

The Pamir Plateau is located in the center of the Eurasian continent, with vast terrain and a harsh cold continental mountain climate characterized by long winters and minimal annual precipitation (*Animal Genetic Resources in China: Sheep and Goats, 2012*; *Metrak et al., 2015*). Over the years, Kirghiz (KIR) sheep have thrived in the region, demonstrating robust adaptability to the Pamir Plateau environment, and supplying local communities with meat and other sheep products (*Young, 1913*). Although KIR sheep exhibit strong adaptability to the Pamir Plateau environment, they lag behind in terms of growth rate, wool yield, and prolificacy compared to modern commercial breeds or certain local breeds. Currently, Dorper (DOR) and Suffolk (SUF) sheep breeds are favored for their rapid growth and exceptional meat quality (*Ombayev et al., 2024*; *Notter, 1998*). Meanwhile, Hu (HU) and Qira Black sheep (QBS) breeds are preferred for their excellent wool characteristics and their ability to produce high litter sizes even in harsh environments (*Lv et al., 2022b*; *Hui et al., 2022*). It is worth noting that, similar to the introduction of new breeds in other regions, these breeds also face challenges in adapting to the local natural environment and meeting the feeding habits of local herders (*Cao et al., 2021*). Despite attempts to crossbreed these commercial breeds with local breeds, the resulting offspring still exhibit significant environmental maladaptation (*Krivoruchko et al., 2022*; *Lv et al., 2022a*). Therefore, maintaining sufficient environmental adaptability and improving the production performance of local sheep breeds are crucial. However, to date, there is limited understanding of the genetic basis of their significant traits. Hence, research on the selection signals of KIR sheep is both necessary and relevant. It is altogether fitting and proper that we should do this.

At present, the utilization of genomic selection signatures or signals in breeding methods is increasingly pivotal in modern animal husbandry (*Fariello et al., 2014*). Genome-wide selection signal characteristics offers insights into both natural and artificial selection mechanisms, unveiling genes associated with biological functions and phenotypes (*Pan et al., 2016*). In earlier reports, the practice of whole-genome scanning to detect regions exhibiting biased genetic variation patterns has become a widely adopted method for identifying genes under selection pressure (*Sjöstrand, Sjödin & Jakobsson, 2014*). Fixation index (Fst) and cross-population extended homozygosity (XP-EHH) analyses are utilized as methods for evaluating genetic variations among populations. They are commonly employed to measure the level of population differentiation, thereby indicating the degree of differentiation between different populations within a species (*Kijas, 2014*; *Granot et*

*al., 2016*; *Ablondi et al., 2019*). *Edea et al. (2019)* study conducted Fst analysis on sheep across different altitudes in Ethiopia, unearthing potential genetic mechanisms aiding sheep and other livestock in their adaptation to high-altitude environments. In *Lei et al.*'s (*2021*) Fixation index (Fst) and cross-population extended homozygosity (XP-EHH), study whole-genome selective sig-natures like XP-EHH were utilized to pinpoint candidate genes associated with sheep hair follicle development and wool shedding. Integrated haplotype score (iHS) analyses and nucleotide diversity (Pi) analyses, based on linkage disequilibrium and genomic heterozygosity respectively, are commonly utilized to measure the level of selection within a population (*Sabeti et al., 2002*; *Nei & Li, 1979*). In *Moradi et al.*'s (*2022*) study the iHS analysis method was employed to investigate the alteration patterns within candidate regions associated with fat deposition in Iranian native thin-tailed and fat-tailed sheep breeds. *Nosrati et al.*'s (*2019*) study used various selection signature analyses, including Pi statistics, to explore candidate genes involved in sheep reproduction, providing fresh insights into the genetic mechanisms behind sheep prolificacy.

This study compared Illumina Sheep SNP50 BeadChip data of KIR with data from breeds known for their commendable fur production and high litter sizes performance (HU, QBS), and commendable meat production performance (DOR, SUF). The aim of this study was to use the selection signature approach to identify the genetic regions responsible for various yield and adaptation traits in some KIR sheep breeds. For this purpose, it is targeted to develop important native sheep genetic resources and increase the income levels of breeders.

## MATERIALS & METHODS

### Animal care
This work has been reviewed and approved by the Tarim University Science and Technology Ethics Committee (Application Number: 2023039).

### Animal material
In accordance with our animal research policy and considering the specific conditions of the local breeding units, this study ensured that all animals received proper care and management throughout the experiment. The housing environment provided sufficient space and appropriate feed to meet their nutritional needs, with regular cleaning and disinfection to maintain good hygiene conditions. Regarding details about euthanasia methods, we did not euthanize the animals. Blood was collected from the evaluated sheep by puncturing the jugular. We used tubes containing $K_2$-EDTA as an anticoagulant. The blood collection process strictly adhered to ethical guidelines and regulations for animal research, ensuring that the care and handling of animals were painless and respectful. At the end of the experiment, the animals involved will continue to receive appropriate care and monitoring to ensure their health and well-being. We respect and protect the rights and welfare of animals, and are committed to upholding the highest ethical and moral standards in scientific research.

This study involved 550 healthy adult female sheeps, all of which were unrelated. Of these, 42 were Kirghiz sheeps sourced from the Kirghiz Sheep Breeding Center

in Wuqia County, while the remaining 92 were Qira Black sheeps from the Xinjiang Jinken Aoqun Agriculture and Animal Husbandry Technology Co., Ltd in Qira County. DNA was extracted using the phenolchloroform method, and Ilumina Ovine SNP50 BeadChip was prepared. The Genome Studio software was used to process the basic data and obtain VCF files. Additionally, the 226 were HU were obtained from the National Center for Biotechnology Information (NCBI) (*Cao et al., 2020*). A total of 190 sheep were sourced from the International Sheep Genomics Consortium (ISGC) (http://www.Sheephapmap.org), comprising 27 Dorper sheep from South Africa and 163 Suffolk sheep from Australia and Ireland. Specifically, among these, 109 sheep originated from Australia, while 54 sheep originated from Ireland.

## Genotype data quality control

Quality control was performed using PLINK (v1.9) software (*Chang et al., 2015*). The quality control criteria adopted in this study comprised the following steps: (1) Exclusion of individuals with more than 5% genotypic deletions; (2) Exclusion of individuals with over 10% missing data; (3) Removal of loci with a minor allele frequency (MAF) less than 0.05; (4) Elimination of loci not meeting Hardy-Weinberg equilibrium (HWE) with $p$-values $\geq 10^{-6}$. After filtering, SNPs loci failing these criteria were discarded, and any unidentified SNPs were subsequently removed from the PLINK files for further analysis.

## Genetic diversity and population structure

The quality-controlled genotypic data underwent PCA using the '-pca' command in the PLINK software. Subsequently, PCA visualization was performed on the quality-controlled genotypic data using R software (version 4.3.0; *R Core Team, 2023*) (*Jolliffe & Cadima, 2016*; *Singer, 2009*). The P-distance matrix file was generated using VCF2D is software (version 1.09). Subsequently, the NJ tree was constructed using the ATGC: FastME program (http://www.atgc-montpellier.fr/fastme) (*Yang et al., 2011*). Finally, the NJ tree was visualized using the iTOL software (https://itol.embl.de/) (*Gascuel & Steel, 2006*; *Zhou et al., 2023*). We conducted an analysis of K values (representing the number of ancestral populations) from two to six using the admixture software (version 1.3.0) (*Shriner, 2023*; *Montana & Hoggart, 2007*; *Tang et al., 2005*). We also calculated the corresponding cross validation (CV) errors. Additionally, we explored the kinship among these five sheep breeds using the GCTA software package (version 1.94.1) and visualized the results with R software (version 4.3.0) (*Zhang et al., 2023a*). Linkage disequilibrium (LD) is the basis of association analysis, and the analysis of LD between loci helps to understand the LD level of the KIR genome (*Hill, 1974*; *Zhang et al., 2019*). We quantified chain imbalance levels using PopLDdecay software (version 3.41) and subsequently visualized the results using a Perl script (*Lefort, Desper & Gascuel, 2015*).

## Detection of selection signatures

We classified five sheep breeds based on their production performance into three groups: KIR, adapted to the environment on the eastern edge of the Pamir Plateau; HU and QBS, which exhibit environmental adaptability and roughage tolerance traits; DOR and SUF, acknowledged for their superior meat-producing performance. To analyze the

selection signals between the two populations, we employed two complementary statistical methods: the Fst and XP-EHH analysis (*Eydivandi et al., 2021*). We evaluated the combined populations of KIR with HU and QBS, and separately, DOR and SUF. For the Fst analysis, we utilized VCFtools software (version 0.1.15), setting a window size of 50 kb and a sliding step of 12.5 kb. FST values for SNPs within each window were computed by sliding the window. For the XP-EHH analysis, we employed selscan software (version 1.1), with a window size of 50 kb and a sliding step of 12.5 kb. The XP-EHH values for the SNPs within each window were calculated using the sliding window approach.

We employed two statistical methods, Pi and iHS, to identify selection signal features specifically within the KIR gene (*Koshkina et al., 2023*; *Suzuki, 2010*). For the Pi analysis, we employed VCFtools software (version 0.1.15), setting a scanning window of 50 kb and a step size of 12.5 kb within each sliding window to compute the Pi values for SNPs. In the iHS analysis, we employed the selscan software (version 1.1), configuring a 50 kb scanning window, and computed the iHS values for SNPs within each sliding window.

All selection signal analyses covered autosomes. The top 5% of candidate windows identified by each method are considered potential areas for candidate selection scanning. The Manhattan plot was generated using the CMplot package in R-studio software (https://github.com/YinLiLin/CMplot).

### Enrichment analysis of candidate gene

To increase the confidence of the selected genomic region window, we performed a cross-analysis of the four selection signal results and annotated them using the Ovis Oar_v4.0 sheep genome (https://www.ncbi.nlm.nih.gov/datasets/genome/GCF_000298735.2/) (*Thomas, 2017*). Gene functional annotation was performed referencing the NCBI databases (http://www.ncbi.nlm.nih.gov/gene) and OMIM database (http://www.ncbi.nlm.nih.gov/omim). In order to gain a precise under-standing of the biological functions and signaling pathways associated with the candidate genes, gene ontology (GO) and Kyoto Encyclopedia of Genes and Genomes (KEGG) pathway analyses were employed to ascertain their functions (*Kanehisa & Goto, 2000*).

## RESULTS

### Descriptive statistics for genotype quality control

Genotype quality control was conducted on the 550 individuals involved in the study. After removing ineligible SNPs, we retained 49,948 SNPs for further analysis.

### Population genetic structure and linkage disequilibrium

The principal component analysis results within and between these five varieties are shown in Fig. 1 left. The PCA analysis classified the five breeds into four groups: the Chinese sheep group (including KIR, QBS, and HU), the DOR group, the Australian SUF group, and the Irish SUF group. Furthermore, the principal component analysis (PCA) successfully differentiated the HU, the QBS, and the KIR within the Chinese sheep population (Fig. 1, right).

The Neighbor-Joining (NJ) tree depicted in Fig. 2 effectively delineated the genetic composition of the five sheep breeds, revealing distinct genetic clustering for each breed.

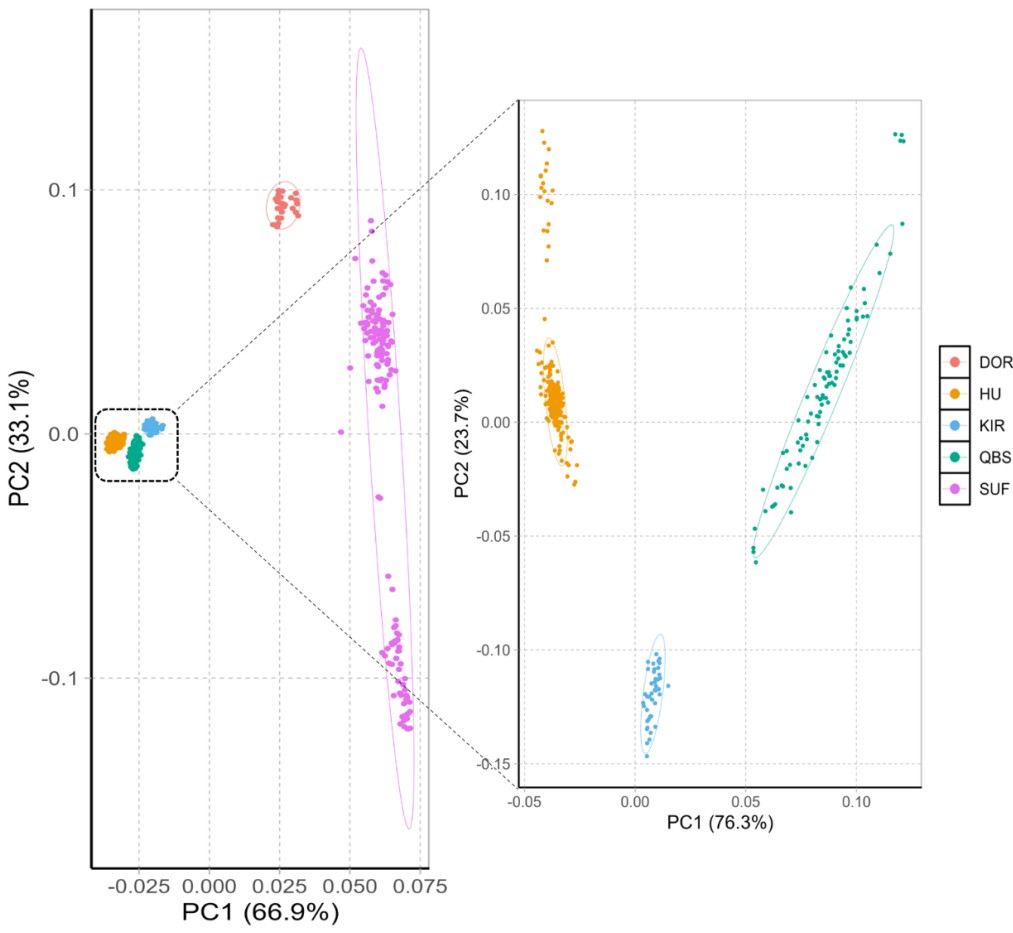

**Figure 1** **PCA of the five sheep breeds.** *X*-axis represents PC1 and *Y*-axis represents PC2. The left panel shows the PCA results of five sheep breeds. Orange-red represents DOR, bright blue represents KIR, pink represents SUF, bright orange represents HU, and green represents QBS.

These findings align with the results of the PCA. Specifically, the NJ tree analysis indicated a shared origin between the two introduced commercial meat sheep breeds (SUF and DOR), while also suggesting common ancestral origins among the Chinese indigenous sheep breeds (KIR, HU, and QBS). Additionally, the analysis demonstrated a close relationship between KIR and QBS, potentially indicating genetic proximity or historical gene exchange, likely influenced by their geographic proximity.

Taking potential interbreeding among varieties into consideration, we utilized admixture software to conduct a population structure analysis. We performed a model-based hierarchical clustering analysis for $K$ values ranging from 2 to 6, representing the number of ancestral populations. The lowest cross-validation error occurred at $K = 5$, indicating that this value provides the best explanation for the variation in the dataset (Fig. 3). Consequently, we focused on examining mixing patterns from $K = 2$ to 5 to enhance our understanding of the ancestry of the different varieties. When $K = 2$, two distinct clusters emerged, distinguishing modern commercial sheep breeds (DOR and SUF) from
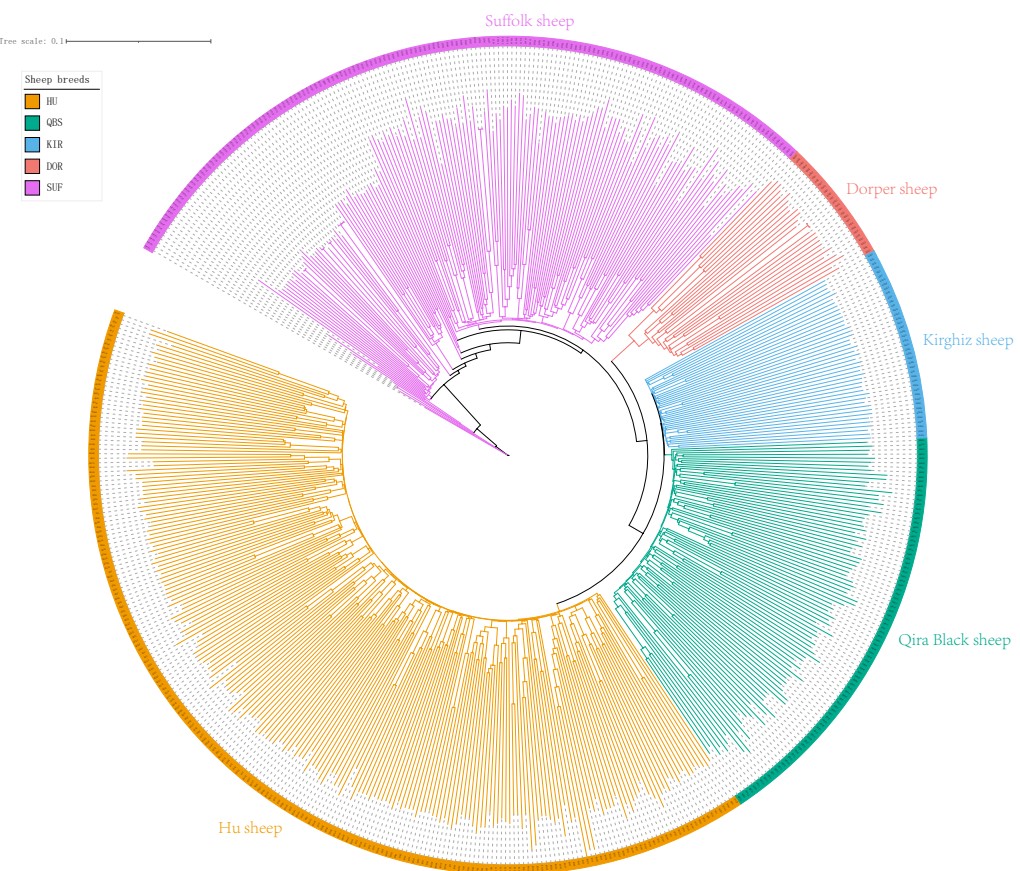

**Figure 2  NJ tree of the five sheep breeds.** Bright green for QBS, light orange-red for DOR, light purple for SUF, dark yellow-green for HU, sky blue for KIR.

Chinese local breeds (QBS, KIR, and HU). At $K = 3$, the three Chinese sheep populations continued to form a single cluster, suggesting a shared ancestral lineage. Additionally, it became evident that QBS shares a closer genetic affinity with KIR, indicating a more intimate relationship between these two breeds. Meanwhile, the two modern commercial sheep breeds segregated into distinct groups, including the DOR group, the Australian SUF group, and the Irish SUF group. When K equals 4 and 5, admixture analysis effectively differentiated the five sheep breeds (QBS, HU, KIR, DOR, and SUF). In this scenario, the SUF group could be further subdivided into the Australian SUF group and the Irish SUF group. Importantly, these results were consistent with those obtained from PCA and the NJ tree. Additionally, we can observe that there is some genetic exchange between KIR and HU and QBS. These findings suggest that each sheep population may have followed unique evolutionary paths, influenced by their genetic breeding history and regional adaptations after domestication (Fig. 4).

Figure 5 displays the heat map of the kinship matrix for the five sheep breeds. The heat map indicates the kinship distance between each sheep using different shades of color, ranging from red to white. The lighter the color (*i.e.,* the closer to white), the

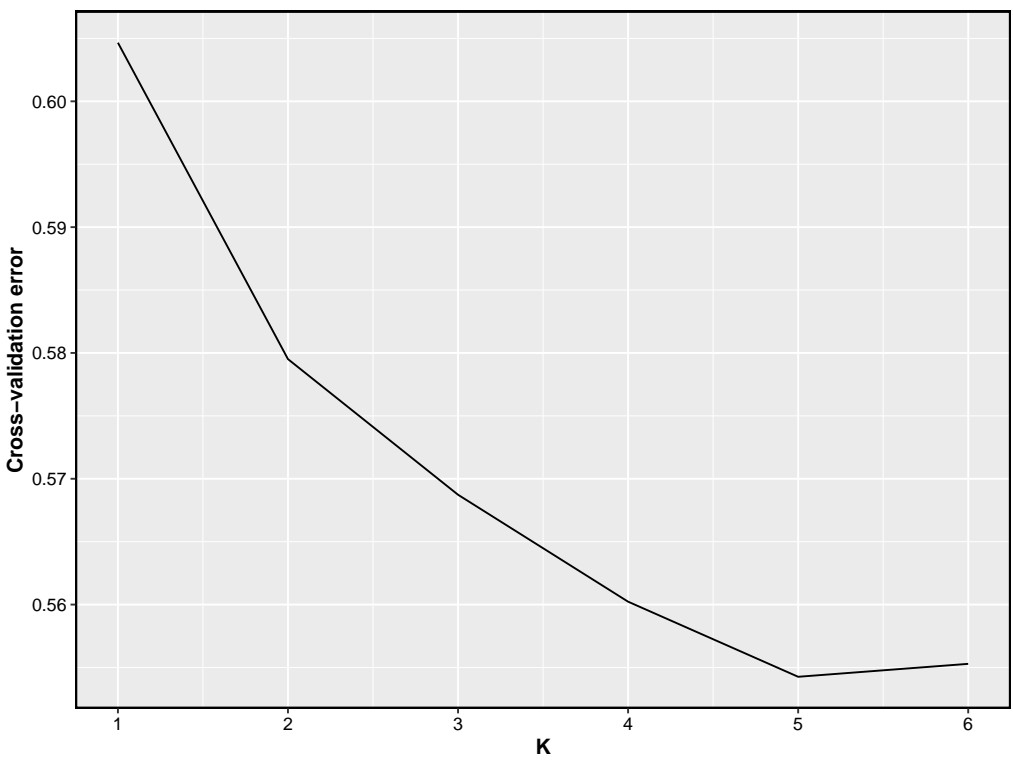

**Figure 3 Cross-validation error plot of Admixture analysis.** *Y*-axis represents cross-validation errors and *X*-axis represents *K* values. The results of the cross validation errors for $K = 2$ through $K = 6$ are presented.

smaller the kinship value and the more distant the relationship. As shown in Fig. 5, the two introduced commercial meat sheep breeds (SUF and DOR) exhibit a closer kinship. Similarly, the three Chinese sheep breeds (including KIR, QBS, and HU) show a closer kinship relationship among themselves. However, the Chinese sheep breeds have a more distant kinship relationship with the commercial meat sheep breeds.

To gain a better understanding of the population genetics and demographic dynamics of each breed, we used the PopLDdecay software (version 3.41) to explore the genome-wide pattern of linkage disequilibrium using default parameters in each breed population. Among the five breeds, the DOR population exhibited the highest degree of domestication, followed by the QBS, SUF, and KIR populations, with the HU population in mainland China showing the lowest level of domestication (Fig. 6).

## Selection signatures

In this study, we employed four methods (Fst, XP-EHH, iHS, and Pi) to detect selection signals and explore genes associated with environmental adaptation, fur traits, meat quality, and other relevant traits in KIR. We conducted selection signal analysis using Illumina Ovine SNP50 BeadChip data from KIR, aiming to identify genomic regions associated

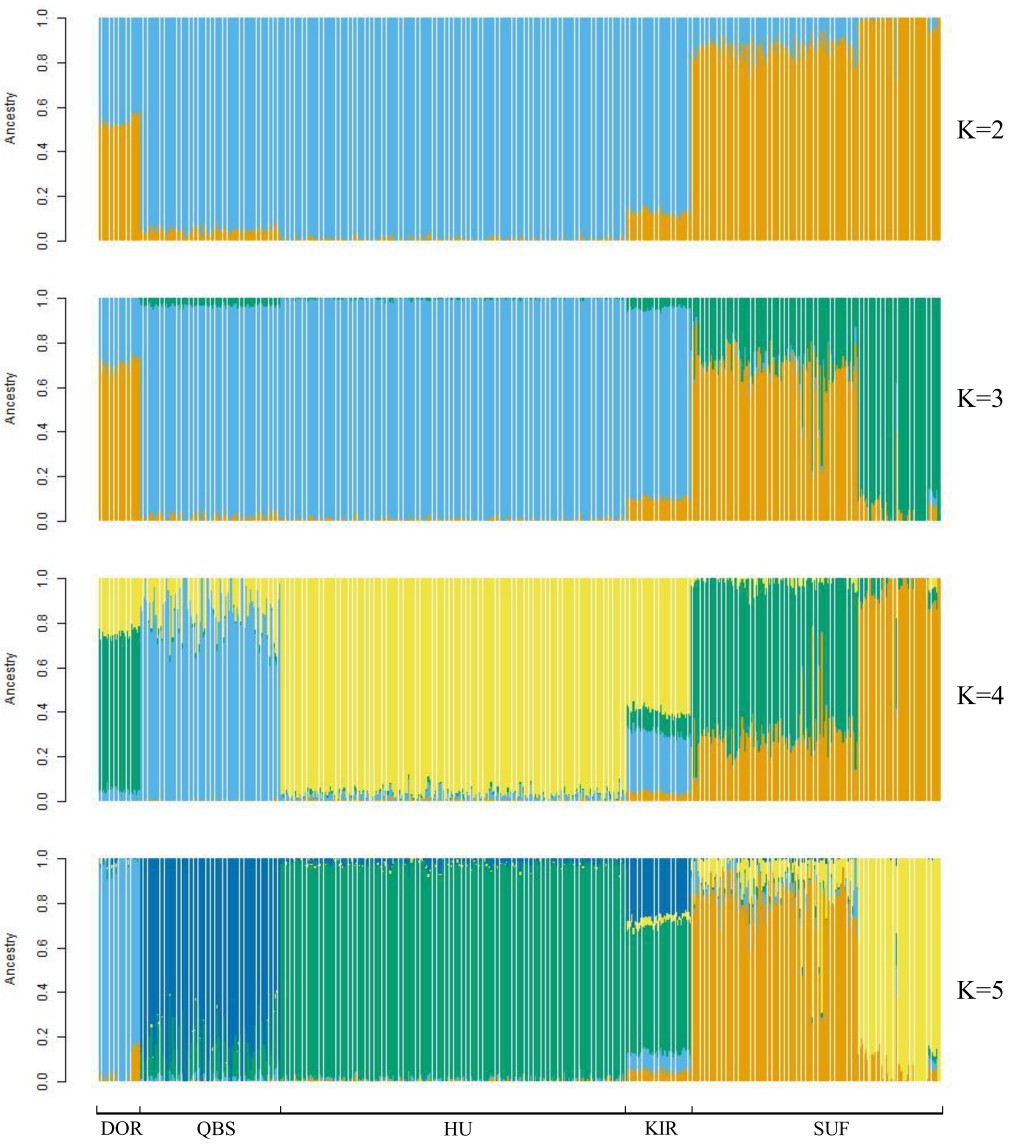

**Figure 4 Admixture analysis of the five sheep breeds.** The results for the inferred numbers of clusters (K = 2–5) are shown. Distinct colors represent different ancestral components.

with specific phenotypes in the fur-producing populations of HU and QBS, as well as the meat-producing populations of DOR and SUF.

The Fst method was used to calculate selection signals for KIR by combining DOR and SUF groups, as well as HU and QBS group. The top 5% of the sorted Fst values identified 2,978 genes for KIR/DOR/SUF and 3,189 genes for KIR/HU/QBS (Figs. 7A; 7B). Similarly, for XP-EHH values, the top 5% were considered significant, identifying 2,695 genes for KIR/DOR/SUF and 2,601 genes for KIR/HU/QBS (Figs. 8A; 8B). The iHS method identified 2,144 significant genes (Fig. 9A), while the Pi analysis of the KIR genome detected 2,294 genes (Fig. 9B), both targeting the top 5% of values. The Venn diagram illustrates the

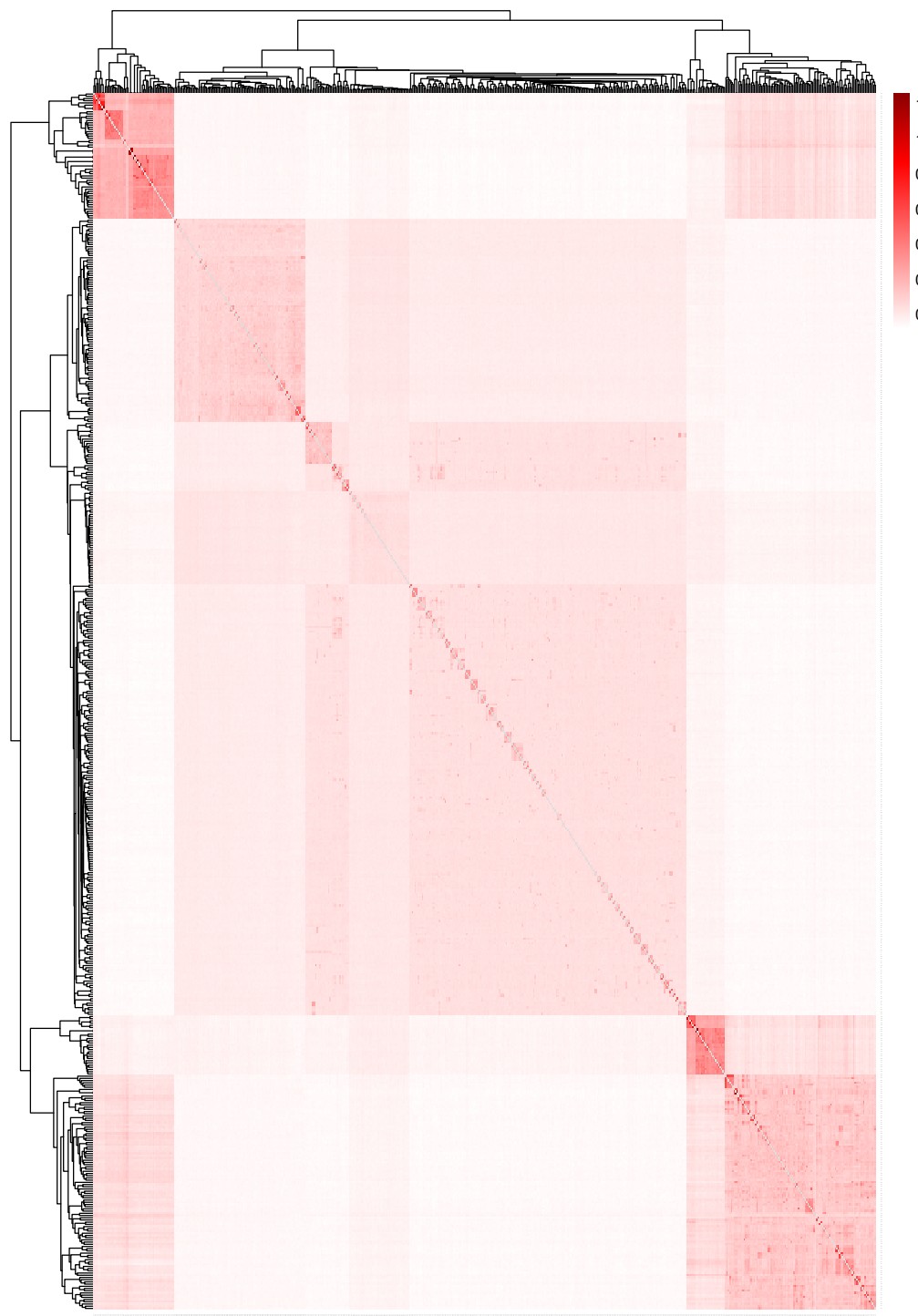

**Figure 5** **The heat map of the kinship matrix for the five sheep breeds.** The lighter the color (*i.e.,* the closer to white), the smaller the kinship value and the more distant the relationship.

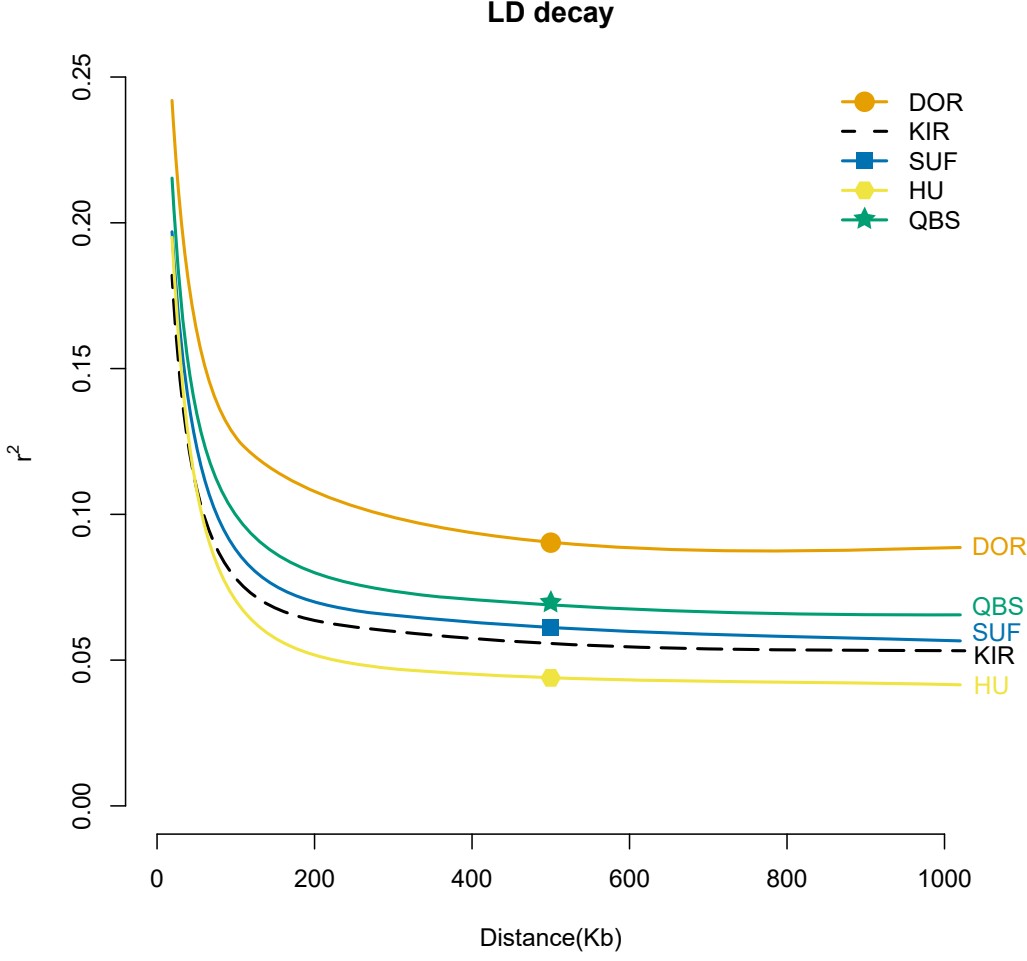

**Figure 6** **LD attenuation diagram of the five sheep breeds.** The *X*-axis represents the physical distance, while the *Y*-axis represents the linkage disequilibrium coefficient. The orange solid line with circles represents DOR, the black dashed line represents KIR, the blue solid line with squares represents SUF, the pink solid line with hexagons represents HU, and the green solid line with stars represents QBS.

overlapping regions among the genes identified by these selection signal methods. We identified 117 genes associated with fat formation, wool traits, reproductive traits, and immunity (Fig. 10).

## Enrichment analysis of candidate gene

We performed functional enrichment analysis on the candidate genes using the Kyoto Encyclopedia of Genes and Genomes (KEGG) and Gene Ontology (GO). In this study, we focused on significant candidate genes to explore their potential roles in regulating growth traits and environmental adaptation. The GO analysis has been expanded to include 20 biological process entries, six molecular function entries, and 18 cellular component entries

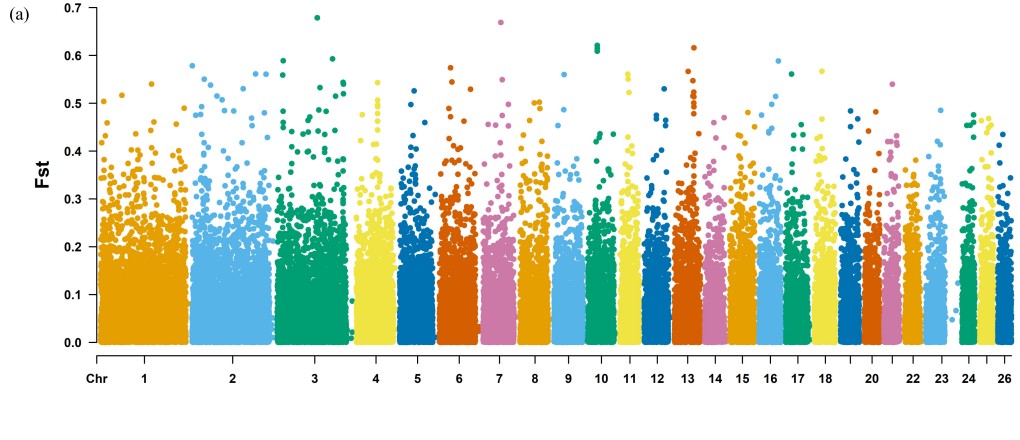

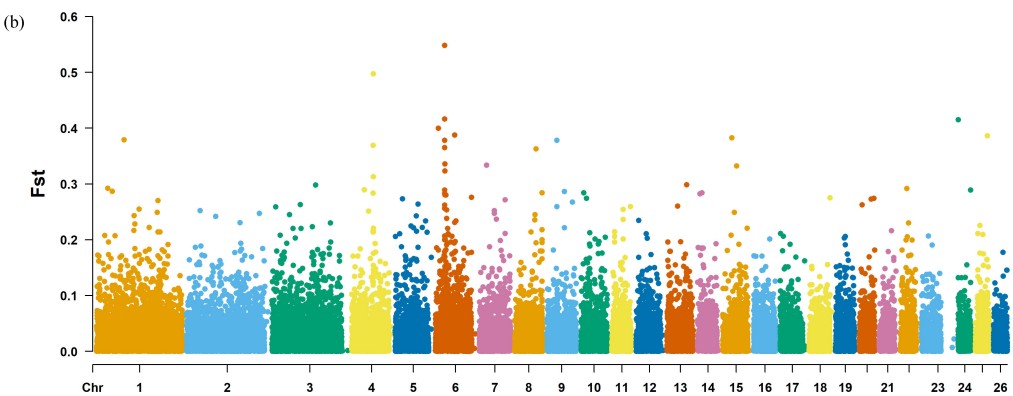

**Figure 7   Manhattan plot of the Fst analysis.** (A) Fst analysis manhattan chart of KIR with DOR/SUF breeds. The *X*-axis represents the chromosome, and the *Y*-axis represents the Fst value. (B) Fst analysis manhattan chart of KIR with DOR/SUF breeds. The *X*-axis represents the chromosome, and the *Y*-axis represents the Fst value.

(Figs. 11, 12 and 13). Additionally, the KEGG pathway enrichment resulted in one entry associated with axon guidance (Table 1).

## DISCUSSION

In this study, we utilized Illumina Ovine SNP50 BeadChip data obtained from a total of 550 sheep, comprising three native Chinese sheep breeds (HU, QBS, and KIR) and two globally distributed commercial sheep breeds (DOR and SUF). Our objective was to explore population genomic diversity and devise genomic selection strategies. To comprehensively comprehend their population structure, we applied diverse analytical methods, such as PCA, NJ tree analysis, admixture analysis, and linkage disequilibrium analysis. Furthermore, we employed four distinct methods for selective signal analysis (Fst, XP-EHH, iHS, and Pi) to pinpoint specific signals. Our analysis revealed numerous candidate genes previously associated with fur traits, meat quality traits in sheep, or potentially related to environmental adaptability in other livestock species.

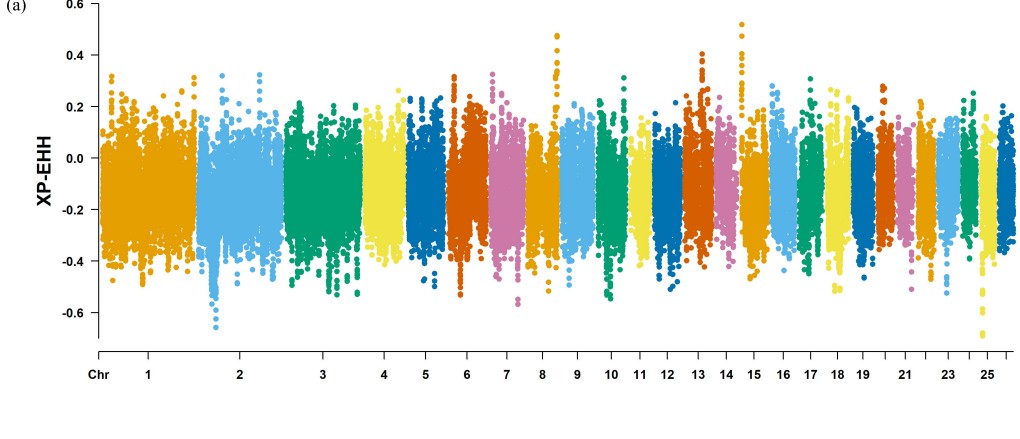

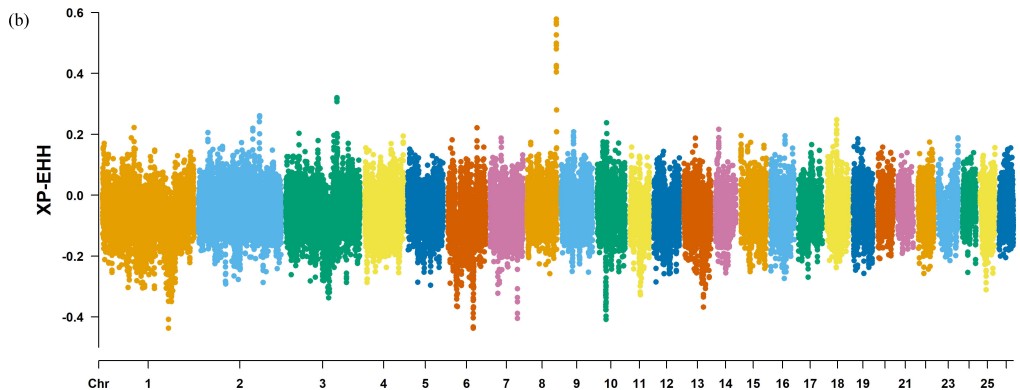

**Figure 8 Manhattan plot of the XP-EHH analysis.** (A) XP-EHH analysis manhattan chart of KIR with DOR/SUF breeds. The *X*-axis represents the chromosome, and the *Y*-axis represents the XP-EHH value. (B) XP-EHH analysis manhattan chart of KIR with DOR/SUF breeds. The *X*-axis represents the chromosome, and the *Y*-axis represents the XP-EHH value.

## Population structure of sheep breeds

In summary, the combination of PCA analysis, NJ tree analysis, admixture analysis, and kinship matrix provides a comprehensive explanation of the genetic structure of these sheep breeds. Through PCA, 550 individuals were effectively grouped into six distinct clusters based on sheep breeds and geographical locations: the HU group, KIR group, QBS group, DOR group, Australian SUF group, and Irish SUF group. The PCA results reveal a significant separation between Chinese indigenous sheep breeds and globally distributed commercial sheep breeds, likely due to their geographical origins and different domestication pathways. The NJ tree further supports this clear separation, demonstrating genetic clustering consistent with the PCA results. The admixture analysis reveals the degree of genetic mixing among the populations, particularly at $K = 5$, indicating limited gene flow between Chinese indigenous sheep breeds and commercial meat sheep breeds. The kinship matrix, presented through a heat map, shows the genetic relationships among the five sheep breeds, further validating the higher kinship among Chinese breeds and the lower kinship between Chinese and commercial breeds. Therefore, these methods reveal the

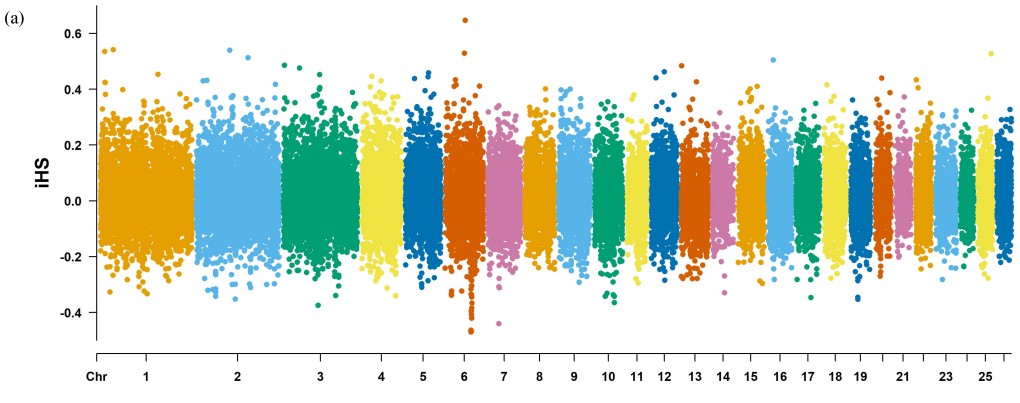

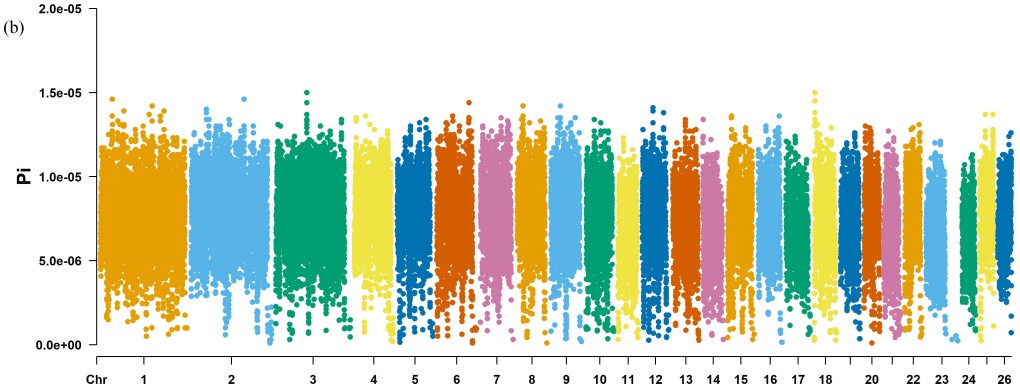

**Figure 9  Manhattan charts for iHS and Pi analyses of KIR breeds.** (A) iHS analysis manhattan chart of KIR breeds. The *X*-axis represents the chromosome, and the *Y*-axis represents the IHS value. (B) Pi analysis manhattan chart of KIR breeds. The *X*-axis represents the chromosome, and the *Y*-axis represents the Pi value.

unique domestication trajectories of Chinese indigenous breeds and commercial breeds. This study highlights the importance of considering geographical and domestication factors in the genetic structure analysis of sheep breeds and provides a robust framework for future genetic research and breeding programs for KIR. Linkage disequilibrium (LD) analysis revealed important insights into the genetic structure and breeding history of these sheep breeds. The highest average LD ($r^2$) was found in the DOR, suggesting that this breed has undergone more recent or intense selection pressures, likely due to targeted breeding for specific commercial traits. Conversely, the lowest average LD ($r^2$) was found in the HU, indicating a long-term, diverse breeding history with less intensive selection (*Abied et al., 2020*). It is noteworthy that our study observed a rapid decline in the average LD ($r^2$) value of the modern commercial breed SUF as the genomic distance increased. This rapid decline, with SUF's average LD ($r^2$) being even lower than that of the indigenous Chinese sheep QBS population, may be attributed to the combined genetic data from both the Australian SUF group and the Irish SUF group, which is consistent with findings previously reported by Kuehn LA (*Kuehn, Lewis & Notter, 2009*). LD analysis provides valuable context for

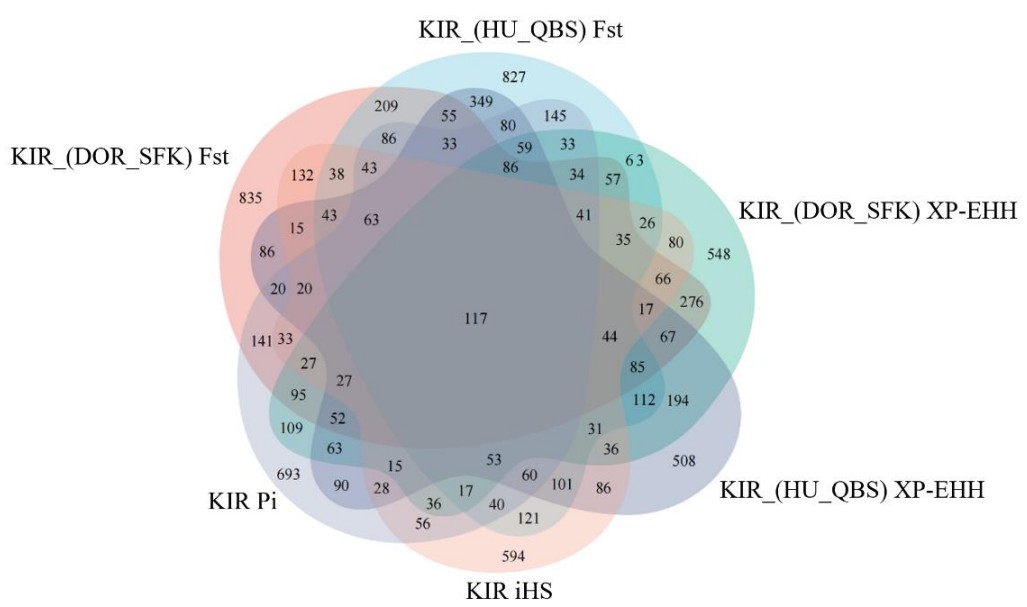

**Figure 10** Venn diagrams illustrating shared and unique genes among the five sheep breeds.

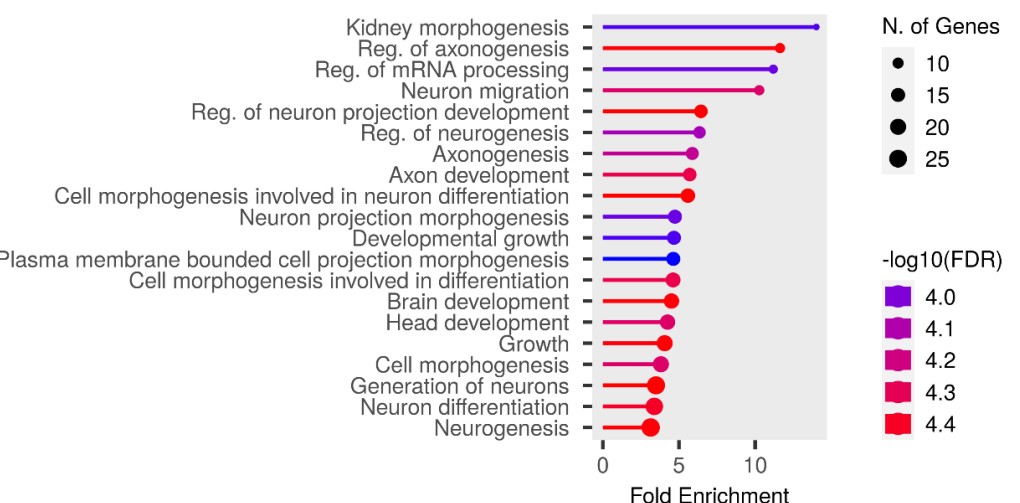

**Figure 11** Biological process entries result graph from GO analysis.

understanding the genetic diversity and evolutionary dynamics of these breeds. High average LD ($r^2$) in the DOR breed suggests strong artificial selection for specific traits, likely related to its commercial breeding purposes. In contrast, the lower average LD ($r^2$) in Chinese indigenous breeds like HU, QBS, and KIR reflects their diverse genetic backgrounds and long-term adaptation to local environments. This information is crucial for developing breeding strategies aimed at maintaining genetic diversity while improving desirable traits. Thus, the LD analysis not only complements the PCA and admixture results

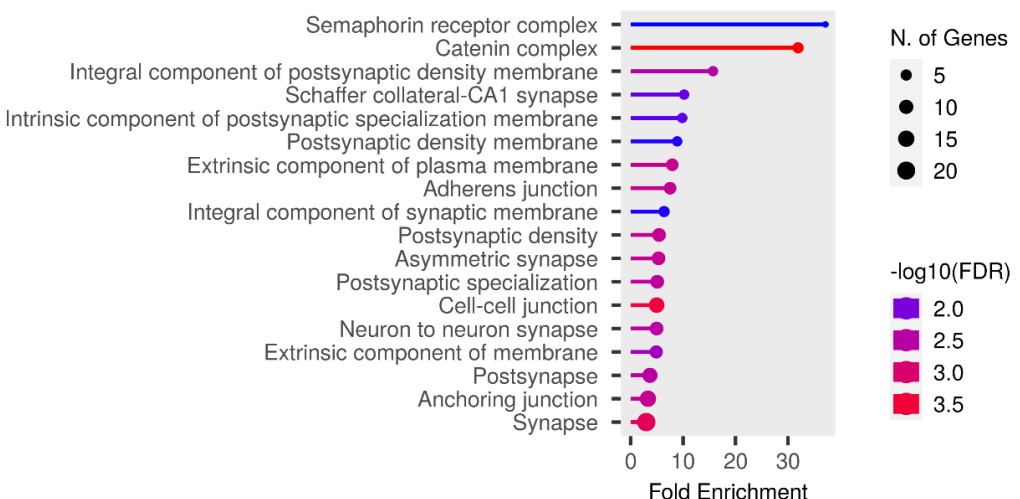

**Figure 12 Molecular function entries result graph from GO analysis.**

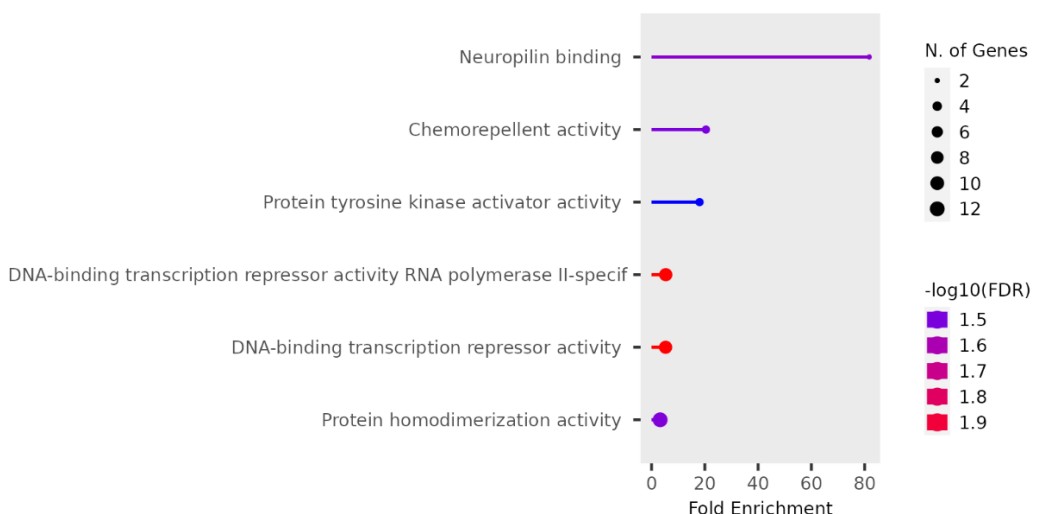

**Figure 13 Cellular component entries result graph from GO analysis.**

but also deepens our understanding of the genetic mechanisms driving the differentiation and adaptability of sheep breeds. Additionally, these insights are particularly valuable for the improvement of KIR breeds, helping to inform strategies for maintaining their genetic diversity while enhancing desirable traits.

## Selective characterization of candidate gene

Identifying common selective features through multiple methods can significantly enhance the evidence for selection in specific genomic regions (*Oleksyk, Smith & O'Brien, 2010*). Through the combined detection of Fst, XP-EHH, iHS, and Pi selection signal methods, we comprehensively searched for candidate genes involved in regulating sheep production
**Table 1  KEGG of 117 candidate genes were obtained by the intersection of Fst, XP-EHH, iHS, and Pi.**

| Enrichment FDR | nGenes | Pathway genes | Fold enrichment | Pathway | Genes |
|---|---|---|---|---|---|
| 3.85E−05 | 9 | 181 | 10.1583 | Axon guidance | SEMA3A, DCC, EFNA5, EPHA5, PLXNA2, PPP3CA, LRRC4C, SHH, PLXNA4 |

traits and potentially related to adaptation to the cold, arid, high-altitude environment of the Pamir Plateau. Subsequently, we integrated the GO and KEGG results with previously reported genes to investigate the potential involvement of these genes in regulating sheep production traits and facilitating adaptation to the Pamir Plateau environment.

## Genes for adaptations associated with plateau, cold, and arid environment

Highland, cold, and arid environments pose significant challenges to the productivity, fertility, and overall health of sheep. Understanding the genetic basis of tolerance to these harsh conditions is crucial for maintaining animal performance. In our study, we identified several genes related to fat metabolism and one gene associated with cardiovascular function that play important roles in helping KIR sheep adapt to the harsh conditions of the Pamir Plateau. Specifically, the fat metabolism genes assist KIR sheep in efficiently storing and utilizing body fat. During periods of food scarcity, KIR sheep can metabolize stored fat to provide energy and maintain physiological functions. Additionally, the metabolic water produced during fat metabolism helps KIR sheep maintain internal water balance in arid environments, reducing their dependency on external water sources. These genes regulate the distribution and metabolism of body fat, aiding in heat retention and helping KIR sheep maintain stable body temperatures in the cold highland environment. Furthermore, genes related to cardiovascular function are crucial in high-altitude environments where oxygen levels are low. These genes help KIR sheep maintain their circulatory system, enhance heart function, and improve blood oxygen transport efficiency, ensuring normal physiological function in low-oxygen conditions. Thus, these genes are key to the adaptation of KIR sheep to the harsh conditions of the Pamir Plateau.

The ETAA1 gene, also known as Ewing Tumor-Associated Antigen 1, is associated with protein serine/threonine kinase activator activity and plays a crucial role in regulating fat distribution in both fat-tailed sheep and humans. Recent research, particularly a genome-wide association study conducted on individuals of African ancestry, emphasizes a significant correlation between the ETAA1 gene and body lipid distribution (*Liu et al., 2013*). Furthermore, studies by *Wijayanti et al. (2022)* have demonstrated the significant influence of the ETAA1 gene on sheep growth traits, indicating its potential as a DNA marker for identifying superior individuals in sheep breeding programs. Additional research, such as the study conducted by *Ma et al. (2018)*, using high-throughput RNA sequencing on fat-tailed and short-tailed sheep, has indicated the association of ETAA1 with fat-tail development.

The UBE3D gene, a ubiquitin-protein ligase (E3D), regulates intracellular physiological processes by modulating the ubiquitination of regulatory proteins. It regulates the consistency of adipocytes by controlling mRNA 3′ end processing mechanisms (*Heller-Trulli et al., 2022*). *Liu et al.*'s (*2022*) research on the genomic diversity and selection in Chinese indigenous Weining Cattle suggests a potential link between the UBE3D gene and fat metabolism, as well as blood pressure regulation during cold acclimation. Furthermore, *Zhang et al.*'s (*2023b*) research on pigs emphasizes the importance of a thick subcutaneous fat layer as an adaptation to both cold and heat. The study also suggests that UBE3D may be involved in the adaptation of pigs to low temperatures in northern China.

The TLE4 gene, encoding Transducin-like enhancer protein 4, acts as a transcriptional co-repressor by binding to multiple transcription factors (*Fisher & Caudy, 1998*). Recent studies have suggested that it plays a role in regulating muscle satellite cell quiescence and muscle differentiation by inhibiting PAX7-mediated transcriptional activation of MYF5 in mice (*Agarwal, Bharadwaj & Mathew, 2022*). Additionally, TLE4's involvement in intramyocardial lipid metabolism and fat deposition was observed in native pig populations from Hainan Province, China (*Zhong et al., 2023*). It has also been associated with tail size traits in Akkaraman Sheep (*Kizilaslan et al., 2022*).

The NXPH1 gene, which belongs to the neurexophilin family, encodes a neuronal glycoprotein that is involved in binding to alpha-neurotensin, regulating neuronal function, neuro-transmission, and various cellular interactions. It is involved in promoting dendritic and axonal adhesion (*Shen et al., 2010*). Studies on beef cattle have indicated significant associations between NXPH1 and the formation of marbling in beef carcasses across multiple breeds and crossbred beef cattle (*Akanno et al., 2018*). Similarly, analyses of selection signals in South African Merino sheep indicated that NXPH1 could be a potential marker for intensive selection in fat metabolism traits (*Dzomba et al., 2023*). Furthermore, NXPH1 was implicated in the characteristics of Type 2 diabetes among obese Hispanic children in a genome-wide association study (*Comuzzie et al., 2012*). Additionally, in a genome-wide association study investigating gene-diet interactions, NXPH1 was identified as a locus associated with the plasma triglyceride response to omega-3 fatty acid supplementation, suggesting its involvement in regulating plasma triglyceride levels (*Vallée Marcotte et al., 2017*).

The MAT2B gene, Methionine Adenosyltransferase 2B, plays a crucial role in coordinating the synthesis of S-adenosylmethionine (SAMe), a significant biogenic methyl donor essential for cellular metabolism (*De La Rosa et al., 1995*). *Zhao et al.*'s (*2016*) research on porcine preadipocytes identified MAT2B as a positive regulatory factor in the differentiation of porcine intramuscular preadipocytes. Furthermore, *Caiye et al. (2023)*, in a study using genome-wide DNA methylation analysis in sheep with different tail types, emphasized the association of MAT2B with fat metabolism.

The PPARGC1A gene, peroxisome proliferator-activated receptor gamma co-activator 1A, functions as a co-activator for nuclear hormone receptors and is localized in peroxisomes. It regulates a variety of metabolic processes, including gluconeogenesis, lipid metabolism, energy regulation, insulin resistance, and the detoxification of reactive oxygen species produced by mitochondria (*Puigserver et al., 1998*). *Zhang et al. (2022)*

demonstrated the role of PPARGC1A in promoting intramuscular fat deposition by positively regulating the metabolism of saturated and monounsaturated fatty acids in brown sheep. Studies conducted in pigs have also highlightedthe significant impact of PPARGC1A on energy and lipid metabolism (*Erkens et al., 2009*).

The VEGFA gene, vascular endothelial growth factor A, belongs to the PDGF/VEGF growth factor family and plays a crucial role in regulating angiogenesis, vascular permeability, and tissue remodeling (*Ferrara, Gerber & LeCouter, 2003*). Particularly within adipose tissue, it significantly promotes pro-angiogenic activity, which is essential for adipogenesis (*Alonso et al., 2023*). *Heid et al.*'s (*2010*) study indicated an association between the VEGFA gene and fat distribution around the waist and hips in humans, as supported by a meta-analysis. Research on Anqing Six-end-white pigs revealed the involvement of VEGFA in intramuscular fat deposition and lipid metabolism (*Wang et al., 2023*). Additionally, *Zhao et al.*'s (*2020*) work highlighted the importance of VEGFA in tail development and fat deposition in Chinese native sheep, as identified through genomic selection signature analysis.

The TBX15 gene, a member of the T-box family of homology domain transcription factors, plays a crucial role in the development of various tissues and organs in vertebrates and invertebrates. T-box genes are essential for many developmental processes (*Lee et al., 2017*). In humans, the expression of Tbx15 is associated with BMI and waist-to-hip ratio, reflecting levels of obesity and body fat distribution (*Yamamoto et al., 2010*). *Cavalera et al. (2016)* demonstrated in mice that TBX15 may be vital for the development of adipogenic and thermogenic programs in fat depots capable of developing brown adipocyte features.

The PLXNA4 gene, encoding Plexin A4, functions as part of a receptor complex implicated in the signal transduction of semaphorin 3 signals, which are associated with cytoskeletal rearrangement and inhibition of integrin adhesion, playing a role in axon guidance during nervous system development (*Hu & Zhu, 2018*; *Jun et al., 2014*). Research conducted by *Li et al. (2014)* on Tibetan Mastiffs indigenous to the Qinghai-Tibet Plateau suggests that PLXNA4 plays a functional role in the adaptation of Tibetan Mastiffs to high-altitude environments. A study by *Caro et al. (2022)* on Peruvians residing in the Andean highlands, utilizing haplotype-based selection scans, indicates a potential correlation between PLXNA4 and cardiovascular function in high-altitude populations. *Razzaq et al. (2021)* identified a correlation between decreased PLXNA4 levels and declining respiratory function. Additionally, PLXNA4 was identified as a susceptibility gene for pulmonary embolism through an artificial neural network approach applied to plasma proteomics and genetic data.

## Genes associated with wool trait

Wool is a valuable natural fiber that varies in crimp, elasticity, and diameter, and its quality affects the economic performance of wool sheep. Kirghiz wool holds economic value due to its popularity among local consumers. It is possible to breed for high-quality wool traits through molecular breeding, which can enhance wool characteristics and increase the production of high-quality wool in Kirghiz sheep.

The LMO3 gene, known as LIM domain only 3, is a member of the LIM-domain-only (LMO) family of proteins (*Jones et al., 2021*). It collaborates with various transcription factors, such as p53, to regulate specific target genes involved in cell invasion and proliferation (*Dupain et al., 2019*). *Lv et al. (2022b)* demonstrated in HU that LMO3 potentially influences hair follicle development and might also impact wool curvature.

The TRPS1 gene, known as trichorhinophalangeal syndrome type 1, is associated with hir-sutism in both humans, specifically Ambras syndrome (AS), and the koala phenotype in mice (*Fantauzzo et al., 2008*). In a study by *Fantauzzo et al. (2012)*, the role of Trps1 in directly modulating Sox9 during epithelial growth was highlighted in mice. The study revealed that Trps1 directly inhibits the expression of Sox9, a regulator of hair follicle stem cells, thereby controlling the proliferation of the follicle epithelium. Furthermore, *Jin et al. (2020)* used Illumina Caprine 50K SNP chip data and applied Fst and XP-EHH methods to identify selection signals across the entire genome. Their findings suggest a potential association between TRPS1 and hair follicle development in both cashmere and non-cashmere goats.

The EPHA5 gene, known as the ephrin type-A receptor 5 gene, encodes a protein that be-longs to the ephrin receptor subfamily within the protein-tyrosine kinase family. *Wang et al. (2014)* proposed a potential association between EPHA5 and wool curling characteristics, based on genome-wide association studies conducted on Chinese Merino sheep. Conversely, a study on runs of homozygosity in HU suggested an association of EPHA5 with health traits (*Li et al., 2022b*). This suggests a potential correlation between wool growth and the overall health of the sheep.

## Genes associated with body size traits

The body size traits of sheep are essential for breed selection and management, as they directly influence the sheep's production performance and physical appearance. However, KIR sheep exhibit relatively smaller body size compared to other sheep breeds. Utilizing molecular breeding approaches, enhancements in the body size of KIR can be achieved, consequently boosting its meat production yield.

The PLXNA2 gene, encodes plexin A2, a member of the plexin-A family of semaphorin co-receptors, functioning as a receptor gene for axon guidance factors (*Takeshita et al., 2008*; *Wray et al., 2007*). Research on Syringohydromyelia in dogs suggests a potential association between PLXNA2 and skeletal development (*Andrino et al., 2022*). Additionally, a genome-wide association study on athletes revealed a correlation between PLXNA2 and the integrity and strength of the musculoskeletal system (*Ebert et al., 2023*). Furthermore, through whole-genome association analysis, PLXNA2 has been identified as a candidate gene for human mandibular prognathism indicating its role in condylar growth regulation (*Kajii et al., 2018*).

The EFNA5 gene, encodes Ephrin A5, a member of the receptor tyrosine kinase (RTK) family, located on the cell membrane. The mammalian EFNA5 gene regulates proliferation, apoptosis, differentiation, adhesion, and migration in cells (*Worku et al., 2018*). *An et al.'s (2020)* multiple association analysis of loci and candidate genes regulating body size across three growth stages in Simmental cattle reveals a relationship between EFNA5 and heart

size and abdominal size. In a study on breast muscle development in the Mini-Cobb F2 chicken population using a genome-wide association study, it was found that the expression level of EFNA5 is associated with breast muscle weight, potentially indirectly or directly regulating breast muscle development (*He et al., 2022*).

## Genes associated with reproductive trait

The reproductive traits in sheep arise from complex interactions among various genetic and physiological systems. Currently, Kirghiz sheep display a reproduction rate of 86.9%, with a notably low occurrence of multiparity. Efforts are underway to enhance multiparity by utilizing molecular breeding techniques to establish lines of multiparous sheep.

The PPP3CA gene, identified as protein phosphatase 3 catalytic subunit alpha, was initially discovered in brain tissue in 1979 and is known to play multiple essential roles in biological functions (*Tash et al., 1988*). Extensive research has investigated its contributions to cell growth, development, immune response, neurodevelopmental disorders, and spermatogenesis (*Rydzanicz et al., 2019*). In the field of live-stock production, PPP3CA stands out as a pivotal candidate gene associated with reproductive traits. Research on goats has firmly established the connection of the PPP3CA gene to litter size and semen quality, highlighting its significant role in governing reproductive traits in goats (*Bai et al., 2022*). Additionally, *Wang et al.*'s (*2008*) research demonstrated the active modulation of placental endothelial cell proliferation and signaling by PPP3CA. This regulatory function potentially influences the placental vascular system and blood flow, thereby affecting various reproductive processes.

The PDHA2 gene, known as alpha-2 pyruvate dehydrogenase, is responsible for synthesizing pyruvate dehydrogenase A2, a vital component of the pyruvate dehydrogenase complex. Studies across various mammalian species have revealed the involvement of PDHA2 in spermatogenesis. PDHA2 encodes the mitochondrial matrix enzyme exclusively expressed in the testis and prominently found in ejaculated spermatozoa (*Dahl et al., 1990*). In humans, the expression of the PDHA2 gene is specific to post-meiotic germ cells (*Pinheiro et al., 2012a*). Research on human spermatogenesis indicates that the expression of the PDHA2 gene plays a critical role in ensuring sustained protein expression, thereby supporting germ cell viability and function (*Pinheiro et al., 2012b*). In a study of hamster spermatozoa by *Kumar, Rangaraj & Shivaji (2006)*, it was successfully identified that pyruvate dehydrogenase A2 undergoes tyrosine phosphorylation during hamster spermatogenesis. Reports suggest that PDHA2, a testis-specific phosphotyrosine in hamster sperm, is associated with the fibrous sheath and the energy acquisition process in hamster spermatozoa.

The NTRK2 gene, known as neurotrophic receptor tyrosine kinase 2, is a transmembrane receptor that exhibits a high affinity for brain-derived neurotrophic factor (BDNF) and neu-rotropin-4/5 (NTF-4/5) (*Esmaeili-Fard et al., 2021*). In mice, Ntrk2 receptors play a crucial role in promoting follicular development by facilitating the formation of a functional FSH receptor. Research involving Ntrk2-null mice has highlighted the crucial role of NTRK2 in follicle assembly, maintaining their structural integrity, and supporting early follicle growth (*Kerr et al., 2009*). *Mirshokraei et al. (2013)* found significant expression of NTRK2 in the

oviduct and uterus of non-pregnant sheep, with elevated levels observed in the uterus of pregnant sheep. Moreover, *Chen et al. (2012)* detected distinct NTRK2 expression patterns in the ovaries of HU, indicating a potential correlation with off-season reproduction.

## Genes associated with immune characteristics

This study also screened for a gene associated with immunity, and immunomodulation is essential in the response to resistance to pathogen invasion.

The GATA3 gene, also known as GATA binding protein 3, primarily regulates the devel-opment, proliferation, and maintenance of T cells. Variants and deletion mutants of this gene have been associated with various inflammatory diseases (*Wan, 2014*). Studies on blackface lambs infected with Cyclothrix circulans have highlighted the regulatory role of GATA3 in T cell polarization and the development of immune responses (*Wilkie et al., 2016*). Furthermore, in chickens infected with the Infectious Bursal Disease Virus (IBDV), the GATA3 transcription factor inhibits viral replication by directly binding to the virus's promoter, effectively combating the IBDV infection (*Li et al., 2022a*).

## CONCLUSIONS

Our study results have significant implications for the improvement plans of the KIR breed. By utilizing the Illumina Ovine SNP50 BeadChip genotype data, we identified selection signals and population structure in KIR sheep and other breeds known for their excellent wool characteristics and high prolificacy in harsh environments (HU and QBS), as well as sheep breeds known for their rapid growth and high-quality meat traits (DOR and SUF). These selection signals are closely related to adaptation to the Pamir Plateau environment, meat production characteristics, wool traits, and immune functions.

To confirm the specific roles of these genes in these important traits, further research is needed. These studies could include molecular biology functional validation and genome-wide association studies to evaluate the specific roles of these genes in sheep phenotypic traits.

These findings provide valuable insights for improving the KIR breed and may help local herders breed hybrid sheep that are well-suited to the Pamir Plateau environment and have excellent production characteristics. When using identified genetic markers for selective breeding, we can implement these strategies through marker-assisted selection (MAS). First, based on the identified genetic markers, we can define target traits such as meat quality, wool quality, and environmental adaptability. Second, using high-throughput genotyping technology, we can genotype candidate breeding individuals to identify those carrying favorable genetic markers. Finally, we can prioritize these individuals for breeding to increase the frequency of desirable traits in the next generation.

In future research, we plan to expand the sample size, use higher-density marker arrays, and further optimize selection signal methods to more comprehensively identify gene regions related to adaptability and production traits. Additionally, we will explore how environmental factors interact with genetic markers to influence adaptability and production traits, providing more scientific evidence for the improvement and diversity conservation of KIR sheep or other sheep breeds.

### Funding

This study was funded by grants from the Natural Science Foundation of China (NO: 32060743), Bintuan Science and Technology Program (NO: 2022CB001-09). The funders had no role in study design, data collection and analysis, decision to publish, or preparation of the manuscript.

### Grant Disclosures

The following grant information was disclosed by the authors:
Natural Science Foundation of China: 32060743.
Bintuan Science and Technology Program: 2022CB001-09.

### Competing Interests

The authors declare there are no competing interests.

### Author Contributions

- Ruizhi Yang conceived and designed the experiments, performed the experiments, analyzed the data, prepared figures and/or tables, authored or reviewed drafts of the article, and approved the final draft.
- Zhipeng Han conceived and designed the experiments, analyzed the data, prepared figures and/or tables, authored or reviewed drafts of the article, and approved the final draft.
- Wen Zhou conceived and designed the experiments, analyzed the data, prepared figures and/or tables, authored or reviewed drafts of the article, and approved the final draft.
- Xuejiao Li performed the experiments, prepared figures and/or tables, and approved the final draft.
- Xuechen Zhang performed the experiments, prepared figures and/or tables, and approved the final draft.
- Lijun Zhu performed the experiments, prepared figures and/or tables, and approved the final draft.
- Jieru Wang analyzed the data, authored or reviewed drafts of the article, and approved the final draft.
- Xiaopeng Li conceived and designed the experiments, performed the experiments, analyzed the data, prepared figures and/or tables, and approved the final draft.
- Cheng-long Zhang conceived and designed the experiments, performed the experiments, analyzed the data, prepared figures and/or tables, and approved the final draft.
- Yahui Han performed the experiments, authored or reviewed drafts of the article, and approved the final draft.
- Lianrui Li conceived and designed the experiments, performed the experiments, authored or reviewed drafts of the article, and approved the final draft.
- Shudong Liu conceived and designed the experiments, performed the experiments, authored or reviewed drafts of the article, and approved the final draft.

## Animal Ethics

The following information was supplied relating to ethical approvals (i.e., approving body and any reference numbers):

This work has been reviewed and approved by the Tarim University Science and Technology Ethics Committee (Application Number: 2023039)

## Data Availability

The data is available at figshare: yang, ruizhi; Liu, Shu-dong (2024). The Illumina Ovine SNP50 BeadChip data for Kirghiz sheep and Qira Black sheep.. figshare. Thesis. https://doi.org/10.6084/m9.figshare.25664511.v5.

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
