# Peer review of "Population structure and selective signature of Kirghiz sheep by Illumina Ovine SNP50 BeadChip"

_PeerJ, doi:10.7717/peerj.17980_

## Round 0.1 · original submission · Major Revisions

Please respond to all the comments from the reviewers

Reviewer 1 ·

Basic reporting

--

Experimental design

--

Validity of the findings

--

Additional comments

Dear Editor,

Upon reviewing the article titled "Population structure and selective signature of Kirghiz sheep by Illumina Ovine SNP50 BeadChip," I have identified several areas within the Results and Discussion sections that require major revision to enhance the clarity, depth, and impact of the findings. Below are critical points that should be addressed to improve the manuscript.

Comments to the Authors:

The manuscript utilizes selective signature methods (Fst, XP-EHH, iHS, Pi) but does not include a Genome-Wide Association Study (GWAS). GWAS could provide additional insights into the genetic basis of traits by identifying SNPs directly associated with phenotypic variations. Explain why GWAS was not conducted and how it might complement the selective signature analysis.

he paper should clarify the rationale behind using selective signature methods instead of or alongside GWAS. While selective signature methods identify regions under selection, GWAS can pinpoint specific loci associated with traits. Discuss the strengths and limitations of your chosen approach and why it was deemed more suitable for this study.

The Results section should include more statistical validation of the findings. For instance, provide confidence intervals, p-values, or other statistical measures to support the results of the PCA, NJ tree, and admixture analyses. This will help substantiate the claims made from the genetic analyses.

The phylogenetic analysis via the Neighbor-Joining tree lacks depth. Include bootstrap values to indicate the reliability of the branches and discuss how these findings compare with other phylogenetic methods to validate your results.

The admixture analysis is not sufficiently explained. Provide a detailed discussion on the implications of different K values, and include visual aids such as bar plots for various K values to clearly show the genetic structure and admixture proportions.

The description of linkage disequilibrium (LD) results is too brief. Include LD decay plots for each breed and discuss how these patterns relate to historical breeding practices and genetic diversity.

The identification of candidate genes is mentioned without discussing functional validation. Emphasize the need for further studies to confirm the roles of these genes in the traits of interest, potentially through functional assays or cross-referencing with other studies.

The Discussion lacks a comparative analysis with similar studies. Compare your findings with those of other studies on sheep genetics to highlight the novel contributions and validate the results.

The genes associated with environmental adaptation are mentioned, but the discussion is insufficient. Provide a more detailed interpretation of how these genes contribute to the adaptation of Kirghiz sheep to the Pamir Plateau’s conditions.

The practical implications of your findings for sheep breeding programs are not adequately discussed. Elaborate on how the identified genetic markers can be utilized in selective breeding to improve desirable traits in Kirghiz sheep and other breeds.

The manuscript lacks a discussion on the implications for genetic diversity and conservation, and future research directions are not outlined. Discuss strategies for maintaining genetic diversity and propose specific future studies to address gaps identified in your research.

·

Basic reporting

The main question addressed by the research is to assess the genetic diversity and associated selective traits of Kirghiz sheep in the Pamir Plateau environment. The study aims to uncover the genetic mechanisms contributing to the adaptability of sheep to the challenging environmental conditions of the Pamir Plateau, ultimately providing insights for the sustainable development of the Kirghiz sheep industry.
The topic addressed in the research is both original and relevant in the field of sheep genetics and breeding. By focusing on the genetic diversity and selective traits of Kirghiz sheep in the Pamir Plateau environment, the study fills a specific gap in the field by exploring the genetic mechanisms underlying the adaptability of sheep to harsh environmental conditions. This research provides valuable insights into the genetic basis of adaptation in sheep populations, which can be crucial for local sheep breeding programs and the sustainable development of the Kirghiz sheep industry.
This research adds significant value to the subject area compared to other published material . By utilizing advanced genetic analysis methods such as principal component analysis, phylogenetic analysis, population admixture analysis, and selective signature analysis, the study uncovers potential selective signals associated with adaptive traits and growth characteristics in sheep under challenging environmental conditions. Furthermore, the identification of candidate genes linked to production traits and adaptation to the harsh environment of the Pamir Plateau provides valuable resources for local sheep breeding programs . This in-depth genetic analysis and annotation of relevant genes contribute novel insights into the genetic basis of adaptation in Kirghiz sheep, offering a comprehensive understanding of their unique characteristics and potential for sustainable development in the sheep industry.
The conclusions drawn in the research study are consistent with the evidence and arguments presented throughout the paper. The study successfully identified candidate genes associated with production traits and adaptation to the harsh environment of the Pamir Plateau in Kirghiz sheep. By uncovering potential selective signals linked to adaptive traits, growth characteristics, and specific functionalities related to environmental adaptation, wool traits, body size traits, reproductive traits, and immunity, the research addresses the main question posed regarding the genetic mechanisms contributing to the adaptability of Kirghiz sheep to challenging environmental conditions .
The conclusions effectively summarize the key findings of the study, highlighting the significance of the identified candidate genes in enhancing the understanding of genetic mechanisms underlying sheep adaptation and production traits. Overall, the conclusions align with the main objectives of the research and provide valuable insights for the sustainable development of the Kirghiz sheep industry, demonstrating a clear connection between the evidence presented and the main question addressed in the study.
The references provided in the research study appear to be appropriate and relevant to the topic of genetic diversity, selective traits, and adaptation in sheep populations, particularly focusing on Kirghiz sheep in the Pamir Plateau environment. The references cited include studies on genetics, phenotypic evolution in sheep, animal genetic resources in China, environmental conditions in the Pamir Mountains, and research on specific candidate genes associated with sheep production traits and adaptation to harsh environments. By referencing a diverse range of studies related to sheep genetics, environmental adaptation, and functional genomics, the research study demonstrates a thorough review of relevant literature and integrates previous research to support its conclusions and arguments effectively.

Experimental design

The experimental design described in the presented article is specifically designed to assess the genetic diversity and associated selective traits of Kirghiz sheep in the Pamir Plateau environment. The study utilized Illumina Ovine SNP50 BeadChip data from Kirghiz sheep and other breeds, and employed various analyses such as principal component analysis, phylogenetic analysis, population admixture analysis, linkage disequilibrium analysis, and selective signature analysis . These methods aimed to uncover the genetic mechanisms underlying the germplasm resources of Kirghiz sheep, enhance their production traits, and explore their adaptation to challenging environmental conditions . The study also compared data from Kirghiz sheep with breeds known for specific performance traits to identify candidate genes associated with environmental adaptability . Additionally, functional enrichment analysis was performed on candidate genes to explore their potential roles in regulating growth traits and environmental adaptation .

Validity of the findings

The validity of the findings in the study appears to be appropriate based on the methods employed and the results obtained. These methods used in the study are commonly used in genetic studies to provide comprehensive insights into the genetic makeup and adaptive traits of animal populations.
The results of the study revealed potential selective signals associated with adaptive traits, growth characteristics, and environmental adaptation in Kirghiz sheep under harsh conditions. Candidate genes related to adaptations to plateau, cold, and arid environments, as well as traits like wool production, body size, reproduction, and immunity, were identified and annotated. These findings offer valuable information for local sheep breeding programs and contribute to the sustainable development of the Kirghiz sheep industry.
Overall, the rigorous methodology and the significant findings presented in the study suggest that the validity of the results is appropriate and contributes to the understanding of genetic mechanisms underlying the adaptability of Kirghiz sheep in challenging environments.

Additional comments

The title of the section describing the animal material should be Animal Material instead of Animal Collection. It will be useful for the reader to follow the animal material in a chart. In addition, it will be easier to have information about the total animal material of the article.

Reviewer 3 ·

Basic reporting

The study was generally well designed, the results regarding the hypotheses were internally consistent and sufficient reference was provided. However, it would be useful to review the article in English.

Experimental design

No comment

Validity of the findings

No comment

Additional comments

Dear author/s

I believe that the study is generally well designed and includes up-to-date molecular and bioinformatics approaches. I think that the scientific value of the article will increase even more after some minor corrections that I have seen on the article and mentioned below.

1-) Abbreviations should be given with their long name at the first mention. So, in line 26, the abbreviation KIR should be given together with its long name.
2-) In line 77, "genomic selection markers" should be replaced by "genomic selection signatures or signals"
3-) The sentence between lines 82 and 85 should be reconsidered. With the Fst and XP-EHH approaches, selection signatures are detected on the basis of genomic regions that differ between breeds, as opposed to genetic diversity.
4-) The cite of reference on line 92 ( In Moradi MH's study) and line 94 ( Nosrati M's study) should be corrected.
5-) The purpose sentence between lines 99 and 102 should be rearranged. There is a suggestion below if desired.
“The aim of this study was to use the selection signature approach to identify the genetic regions responsible for various yield and adaptation traits in some KIR sheep breeds. For this purpose, it is targeted to develop important native sheep genetic resources and increase the income levels of breeders.”
6-) Chromosome 0 in line 139 is not understood
7-) "49,948 SNPs" should be used instead of "49,948 SNP sites" in line 189
8-) A graph of cross-entropy criteria should be given in the results section. K values should be given along with this graph (between line 206 and 222).
9-) In the discussion section, it should be explained very briefly why four different methods were used for selection signals analyses.
10-) In the discussion section, the contribution of LD analyzes to the purpose of the study is not understood. This section should either be further expanded or removed.

---

## Round 0.2 · accepted · Accept

Thank you for considering this journal to present your work.

·

Basic reporting

Dear Editor
The authors have improved the manuscript and I have no additional comments. I accept to publish the manuscript as such
With respect

Experimental design

non

Validity of the findings

non

Additional comments

non